# Income differences in COVID-19 incidence and severity in Finland among people with foreign and native background: A population-based cohort study of individuals nested within households

Sanni Saarinen[1]*, Heta Moustgaard[1,2], Hanna Remes[1], Riikka Sallinen[1], Pekka Martikainen[1,3,4]

1 Population Research Unit, Faculty of Social Sciences, University of Helsinki, Helsinki, Finland, 2 Helsinki Institute for Social Sciences and Humanities, University of Helsinki, Helsinki, Finland, 3 The Max Planck Institute for Demographic Research, Rostock, Germany, 4 Department of Public Health Sciences, Stockholm University, Stockholm, Sweden

* sanni.e.saarinen@helsinki.fi

**Data Availability Statement:** Due to data protection regulations of the national register-

## Abstract

### Background

Although intrahousehold transmission is a key source of Coronavirus Disease 2019 (COVID-19) infections, studies to date have not analysed socioeconomic risk factors on the household level or household clustering of severe COVID-19. We quantify household income differences and household clustering of COVID-19 incidence and severity.

### Methods and findings

We used register-based cohort data with individual-level linkage across various administrative registers for the total Finnish population living in working-age private households ($N$ = 4,315,342). Incident COVID-19 cases ($N$ = 38,467) were identified from the National Infectious Diseases Register from 1 July 2020 to 22 February 2021. Severe cases ($N$ = 625) were defined as having at least 3 consecutive days of inpatient care with a COVID-19 diagnosis and identified from the Care Register for Health Care between 1 July 2020 and 31 December 2020. We used 2-level logistic regression with individuals nested within households to estimate COVID-19 incidence and case severity among those infected.

Adjusted for age, sex, and regional characteristics, the incidence of COVID-19 was higher (odds ratio [OR] 1.67, 95% CI 1.58 to 1.77, $p$ < 0.001, 28.4% of infections) among individuals in the lowest household income quintile than among those in the highest quintile (18.9%). The difference attenuated (OR 1.23, 1.16 to 1.30, $p$ < 0.001) when controlling for foreign background but not when controlling for other household-level risk factors. In fact, we found a clear income gradient in incidence only among people with foreign background but none among those with native background. The odds of severe illness among those infected were also higher in the lowest income quintile (OR 1.97, 1.52 to 2.56, $p$ < 0.001,

holders providing the data, we are not allowed to make the data available to third parties. Interested researchers have the possibility to obtain data access by contacting the following register-holding public institutions: Statistics Finland (http://www.stat.fi/tup/mikroaineistot/index_en.html), Contact by email tutkijapalvelut@stat.fi Findata (https://findata.fi/en/), Contact by email info@findata.fi.

**Funding:** PM, SS, and RS were supported by the Academy of Finland (#308247, #345219), https://www.aka.fi/en/. PM has also received funding from the European Research Council under the European Union's Horizon 2020 research and innovation programme (grant agreement No 101019329), https://erc.europa.eu/. The study does not necessarily reflect the Commission's views and in no way anticipates the Commission's future policy in this area. Open access funded by Helsinki University Library. The funders had no role in study design, data collection and analysis, decision to publish, or preparation of the manuscript.

**Competing interests:** The authors have declared that no competing interests exist.

**Abbreviations:** COVID-19, Coronavirus Disease 2019; GDPR, General Data Protection Regulation; ICC, intraclass correlation; OECD, Organisation for Economic Co-operation and Development; OR, odds ratio.

28.0% versus 21.6% in the highest quintile), but this difference was fully attenuated (OR 1.08, 0.77 to 1.52, $p = 0.64$) when controlling for other individual-level risk factors—comorbidities, occupational status, and foreign background. Both incidence and severity were strongly clustered within households: Around 77% of the variation in incidence and 20% in severity were attributable to differences between households. The main limitation of our study was that the test uptake for COVID-19 may have differed between population subgroups.

## Conclusions

Low household income appears to be a strong risk factor for both COVID-19 incidence and case severity, but the income differences are largely driven by having foreign background. The strong household clustering of incidence and severity highlights the importance of household context in the prevention and mitigation of COVID-19 outcomes.

## Author summary

### Why was this study done?

- Large body of evidence indicates a higher risk for Coronavirus Disease 2019 (COVID-19) infection, severity, and mortality among people with low socioeconomic position. However, little is known about the reasons for this.

- Furthermore, the quality of the existing evidence is hampered by data limitations such as nonrepresentative samples and area-level measurement of socioeconomic position.

- Even though intrahousehold transmission is a significant source of COVID-19 infections, studies to date have not analysed socioeconomic risk factors at the household-level or the household clustering of severe COVID-19.

### What did the researchers do and find?

- In a population-based cohort study ($n = 4.3$ M) from Finland, we showed that both COVID-19 incidence and case severity are higher in low-income households, but that the income differences are largely driven by other household- and individual-level risk factors.

- The increased risk of COVID-19 infection in low-income households was only present among people with foreign background and nonexistent among those with a native background.

- COVID-19 incidence and case severity are both strongly clustered within households: 77% of variation in incidence and 20% in case severity were attributable to differences between households.

**What do these findings mean?**

- Low household income is not an independent risk factor for COVID-19 outcomes among people with a native background. However, people with foreign background living in low-income households are particularly vulnerable and should be considered for targeted preventive measures.

- The strong household clustering of COVID-19 incidence and severity highlights the importance of the household context in understanding the microlevel dynamics of the COVID-19 pandemic.

## Introduction

Following the outbreak of the Coronavirus Disease 2019 (COVID-19) pandemic, evidence has accumulated concerning the unequal distribution of infections, severity, and mortality across socioeconomic groups [1–14]. Various studies have found a higher COVID-19 incidence among people with low education or low income [1–6], but few of them controlled for other established risk factors, such as household size and composition, occupational exposures, ethnicity, or foreign background [15–18]. It thus remains unclear, whether the higher incidence among people with low socioeconomic position is due to occupational exposures, larger households, or other sociodemographic risk factors that are more common among people with lower socioeconomic position. Higher rates of severe COVID-19 resulting in hospitalization or death have also been reported among these groups [7–9,11]. However, as many studies assess these outcomes among the general population as opposed to the infected [7–9], it remains unclear whether socioeconomic position influences the risk of exposure and infection, or case severity, i.e., outcome once infected, or both. The few previous studies that assess case fatality or mortality among the infected have reported inconsistent findings on the impact of socioeconomic factors [10,11,19]. In fact, most studies on the unequal burden of COVID-19 have not been representative of the general population [15,20].

Another key limitation of the current literature on the socioeconomic differences in COVID-19 outcomes is that COVID-19 risk factors have rarely been assessed on the household level [15], and most studies have relied on area-level socioeconomic measures [1–3,5,6,8–13,19]. The lack of household-level data is a major limitation because both socioeconomic risk factors and poor health tend to cluster in households. People who commute or work in high-risk occupations share the risk with their household members, for example, and the probability of secondary transmission depends on household composition. Furthermore, although multiple studies have shown that intrahousehold transmission is a significant source of new COVID-19 infections [21,22], there are no studies on the household clustering of severe COVID-19 cases. Quantifying the significance of the household context for both incidence and severity enhances our understanding of the microlevel dynamics of the COVID-19 pandemic. This is also a crucial public health issue in that the household clustering of severe illness, particularly among socioeconomically deprived or otherwise vulnerable households, could result in the widening of health inequalities [23].

This study aims to address these limitations of the current literature. Using Finnish total population data on individuals nested within households, we investigate how household income is associated with (1) the risk of COVID-19 infections; and (2) the risk of severe illness

once infected. We use household income as an indicator for the multifaceted concept of socio-economic position. We focus on household income as it is reliably measured and available for all individuals irrespective of their age, employment, or immigrant status. We examine whether the socioeconomic gradient in COVID-19 outcomes found in previous studies is independent of other important COVID-19 risk factors, such as work and school exposures of the household members, household size, foreign background, and comorbidities. We also assess income differences in COVID-19 infections and severity across households of different size and for people with native and foreign background. Furthermore, we quantify the household clustering of COVID-19 infections and severe illness.

## Methods

This study is reported according to the Strengthening the Reporting of Observational Studies in Epidemiology (STROBE) guidelines (S1 Checklist). We did not have a pre-documented analysis plan. The modelling strategy and analyses were planned in spring 2021 and revised according to reviewer feedback.

### Setting and study population

We used individual-level data on the total population of Finland living in private households at the end of 2018, and alive at the end of 2019 (Fig 1), to model the risk of COVID-19 infections from 1 July 2020 to 22 February 2021, and severe illness among those infected from 1 July to 31 December 2020. The different time ranges are due to differences in data availability between sources. Our main analyses are restricted to working age and younger populations, i.e., all members of households with at least 1 person under the age of 65 at the end of 2019. These 1.9 million households comprised 4.3 million people: 76% of them consisted of 1 family or a couple, and most of the rest (17%) were single-person households. The proportion of households with at least 1 member aged 65 or over was 8%.

### Outcomes

Information on laboratory-confirmed COVID-19 cases (ICD-10 code U07.1) was obtained from the National Infectious Diseases Register. As an indicator of severe illness, we used inpatient care lasting at least 3 consecutive days with a primary or secondary diagnosis of COVID-19. Data on inpatient care came from the Care Register for Health Care.

### Household-level characteristics

Annual household income, including all taxable income of all household members in 2018, was divided by the number of consumption units using the Organisation for Economic Co-operation and Development's (OECD) modified equivalence scale and categorized into quintiles. Household size was originally categorized as 1, 2, 3, or 4+ based on the number of household members at the end of 2018. We changed the categorization to 1, 2, 3, 4, or 5+ following a peer reviewer's observation that the 4+ class could be further divided.

People are exposed to COVID-19 infection not only through their own, but also their household members' social contacts. We included several indicators that reflect such indirect exposures to social contacts at work, school, and daycare. The indicators are based on the household members' main occupational activity and the presence and age of children in the household. These dummy variables identified households with at least 1 (1) lower nonmanual employee; (2) self-employed person; (3) manual worker; (4) student in secondary or tertiary education; (5) child aged 13 to 15; (6) child aged 7 to 12; and (7) child aged below the age of 7.

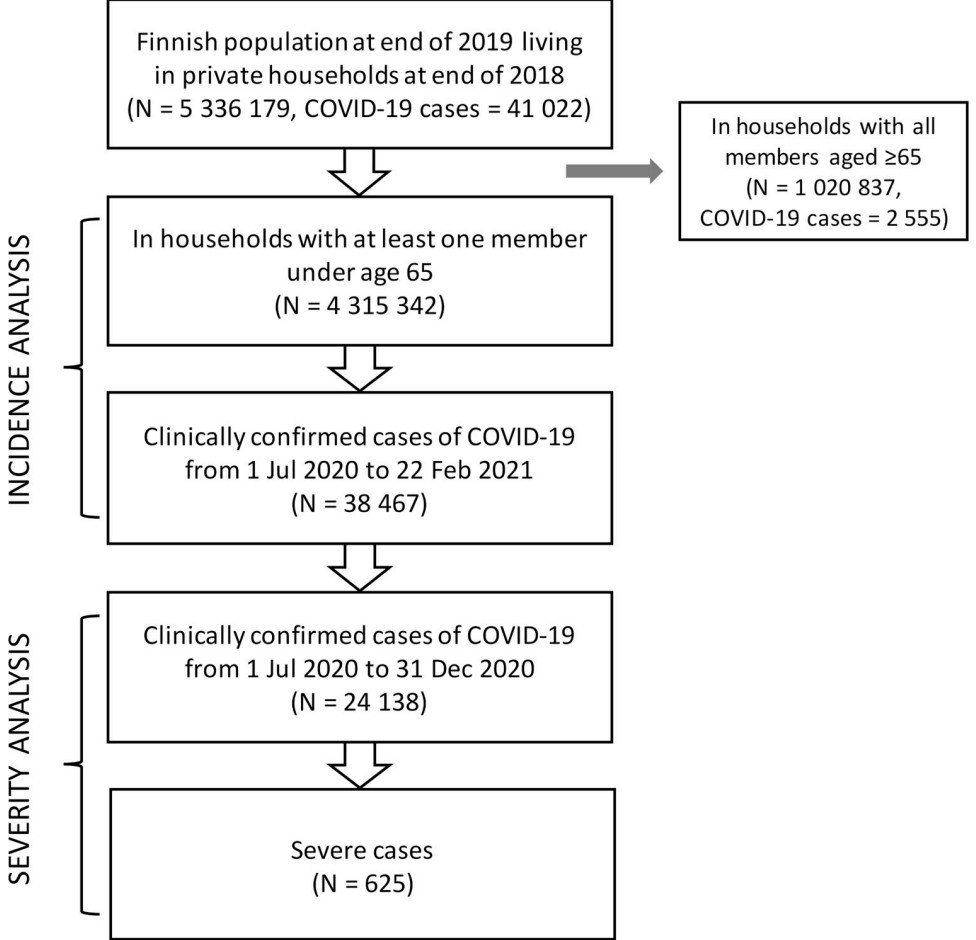

**Fig 1. Data extraction flow chart.** We obtained individual- and household-level data on demographic and socioeconomic characteristics from the population registers maintained by Statistics Finland, and measures of COVID-19 incidence and case severity from registers maintained by the Finnish Institute for Health and Welfare. The national health care registers cover both public and private sector health care providers. The individual-level data were linked using personal identification codes assigned to each resident and household-level data using a unique household identification code. Individuals were nested within households at the end of 2018 because more recent information was unavailable. All the covariates were measured at the most recent time point available in the registers (2017 to 2019).

Information on occupational status was from the year 2017 and student status from the year 2018. The ages of the children in the household were measured at the end of 2019. We based the age categories on educational activity outside the household during the study period: Children in primary education and care (aged under 13) had contact teaching, older children (age 13 to 15) alternated between contact and remote teaching, and most secondary- and higher-level students studied remotely.

The postal code of the permanent place of residence at the end of 2019 was used to define regional characteristics. We categorized urbanicity as: (1) urban area; (2) peri-urban area (including local centres in rural areas); and (3) rural area. We also controlled the analyses for whether the place of residence belonged to the Metropolitan hospital district of Helsinki and Uusimaa (HUS) or to another. HUS is the largest hospital district in Finland and had the highest cumulative number and share of COVID-19 cases compared to the other districts during the whole study period (S1 Table).

### Individual-level characteristics

Sex was measured as binary and age in years at the end of each calendar year.

Foreign background (no/yes) was defined as persons who themselves, both parents, or the only known parent were born outside Finland and whose native language was not Finnish or Swedish.

Comorbidities were identified from information on the right to special reimbursement for medicinal expenses related to specific diagnosed chronic conditions, obtained from the registers of the Social Insurance Institution of Finland. These conditions included cancer, kidney failure, chronic lung disease, diabetes (types 1 and 2), chronic heart disease (heart failure, hypertension, coronary heart disease), and psychotic disorders. Each condition was measured as a separate dummy variable (no/yes), and an individual could have multiple conditions.

Personal occupational status was categorized as: (1) upper nonmanual employee; (2) lower nonmanual employee; (3) self-employed; (4) manual worker; (5) student; (6) pensioner; and (7) other (including unemployed and unknown). This information was measured in 2017, and children aged under 16 were assigned to the same category as the reference person in the household.

### Ethics statement

This study is based on secondary data collected for administrative and statistical purposes, and we have obtained permission to access these data for the purpose of this study from Statistics Finland (permission #TK-53-339-13) and Findata Health and Social Data Permit Authority (permission #THL/2180/14.02.00/2020). Access to these data has been granted after consideration by the ethical boards of these statistical authorities. The study complies with the national legal framework for accessing anonymous personal data for scientific research carried out in public interest. The legal basis is stated in the Finnish Personal Data Act (523/1999), Act on Secondary use of Social and Healthcare data (552/2019), Finnish Statistics Act (280/2004), and the EU General Data Protection Regulation (GDPR). The GDPR permits processing this type of data for research without using the GDPR consent (Art. 9 of the GDPR).

### Statistical analyses

**Analytical strategy.**   First, we present the incidence rates of COVID-19 infection, and severe illness among the infected for people aged under 65 and over 65, categorized by sex and household income. All further analyses are restricted to the households with at least 1 person aged under 65 years. This is because our main interest lies in household-level risk factors for COVID-19 and the variation in income as well as the distribution of other household-level and individual risk factors is very different for the older population. The interpretation of household income also differs between the working age and retired populations. Finally, only 6% of COVID-19 infections during the study period were diagnosed in households with only over 65 year olds.

We use 2-level logistic regression to model the risk of COVID-19 infection and the risk of severe illness due to COVID-19 among those who had a registered infection. Two-level models with individuals (level 1) nested within households (level 2) are needed to account for the non-independence of outcomes among members of the same households. If this correlation is not taken into account, the standard errors will be underestimated, leading to biased statistical inference [24].

Our modelling strategy was guided by our interest to examine whether and to what degree any income differences in COVID-19 outcomes are confounded by other sociodemographic risk factors more commonly found among individuals in lower socioeconomic position. We

first adjusted our models for age, sex, and regional characteristics as basic demographic confounders to obtain a baseline association between household income and COVID-19 outcomes. Then, we adjusted these baseline models with other established COVID-19 risk factors, namely household size, work and school exposures, and foreign background. These factors were included to the baseline model one by one to explicitly assess the potential confounding role of each covariate. In addition, we tested for a potential interaction between income and household size to assess whether the association between household income and COVID-19 outcomes varied across households of different size. In response to reviewer feedback, we further tested for the interaction between household income and foreign background and a 3-way interaction between household income, household size, and foreign background.

The analyses of COVID-19 incidence and severity included partly different covariates (see below for the exact composition of the models). For the analyses of incidence, we focused on household-level risk factors such as household size and work and school exposures of any household member because they affect the infection risk of all household members through secondary transmission. In contrast, as case severity is likely to be more strongly affected by individual-level vulnerability, in the analyses of severe illness we focused instead on individual-level occupational class and comorbidities.

We used Stata version 16.1 to conduct the analyses, with the procedure "melogit" for the multilevel modelling.

**Analyses of incidence.** In Model 1, we assessed the risk for COVID-19 infection by household income controlling for age (linear and squared to account for nonlinear effects), sex, and regional characteristics. We also estimated a similar model—including age, sex, and regional characteristics but excluding household income—separately for each of the other COVID-19 risk factors: household size, each household-level indicators of work and school exposure, and foreign background. These models provide crude baseline associations between each risk factor and COVID-19 incidence. We then built on the first model with household income, separately adjusting for household size in Model 2, for all household-level indicators of work and school exposures in Model 3, for foreign background in Model 4, and finally for all the variables simultaneously in Model 5.

We further modelled the interaction effects of household income with household size and foreign background. We adjusted the first interaction model for age, sex, and regional characteristics (as in Model 1 above), and the second model additionally for all other risk factors (as in Model 5 above). We based the interaction models on 1-level logistic regression with household-clustered standard errors.

**Analyses of severe illness due to COVID-19.** We modelled the risk of severe illness due to COVID-19 among those who had a registered infection. As in the incidence analyses, Model 1 adjusted for age, sex, and regional characteristics and was run separately for household income, household size, each comorbidity, personal occupational status, and foreign background. Model 2 included household income, additionally adjusting for all comorbidities. Subsequent models were built on Model 2, with separate adjustments for household size (Model 3), for personal occupational status (Model 4), for foreign background (Model 5), and finally for all variables simultaneously (Model 6).

As in the incidence analyses, we modelled the interaction effects of household income with household size and foreign background status using 1-level logistic regression with household-clustered standard errors. We adjusted the first interaction model for age, sex, regional characteristics, and comorbidities, and the second model additionally for all other risk factors (as in Model 6 above).

**Clustering within households.** We calculated intraclass correlations (ICC) for the 2-level regression models. ICC was defined as $v/(v + 3.29)$, where $v$ is the between-household variance

[25], and gives is the percentage of total variation in COVID-19 incidence and severity that is attributable to differences between households [25]. It can also be interpreted as the correlation in outcomes between household members [26].

**Sensitivity analyses.** In response to peer reviewers' comments, we implemented 3 sensitivity analyses. First, we reestimated case severity models with the outcome defined as hospitalization of any length with COVID-19 diagnosis. This was done because the length of hospital stay may not in itself be a strong criterion for COVID-19 severity. However, in the main analyses, the indicator of 3+ days in hospital was used to ensure that we capture cases severe enough to warrant continuous inpatient care and exclude very short stays, possibly due to other health conditions. Second, we reestimated the analyses of severe COVID-19 including only primary COVID-19 diagnoses in order to exclude those who were in hospital with a COVID-19 diagnosis but not necessarily because of COVID-19. Finally, in order to make our results more comparable with the previous studies assessing hospitalization and mortality in the full population, we reestimated the severity models for the full <65 population.

## Results

Of the total 41,022 COVID-19 cases registered from 1 July 2020 to 22 February 2021, 94% were diagnosed in people living in under-65 households (i.e., households with at least 1 member aged less than 65). The incidence among both men and women living in these households was clearly highest in the lowest income quintile (around 1,300 per 100,000 versus around 800 in the other quintiles: Fig 2). The COVID-19 incidence was much lower among men and women living in over-65 households (i.e., households with all members aged 65 or over), possibly due to fewer household-level risk factors, and there was little variation by household income.

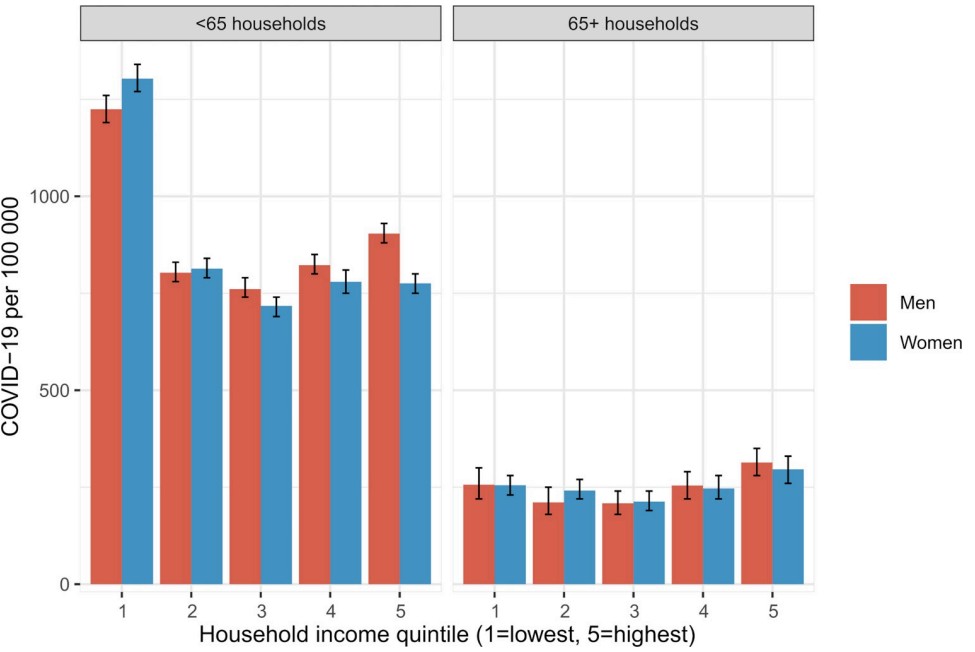

**Fig 2. COVID-19 incidence by sex and household income quintile.** Incidence per 100,000 from 1 July 2020 to 22 February 2021 among individuals living in households with people aged under 65 and over 65. The whiskers represent 95% confidence intervals.

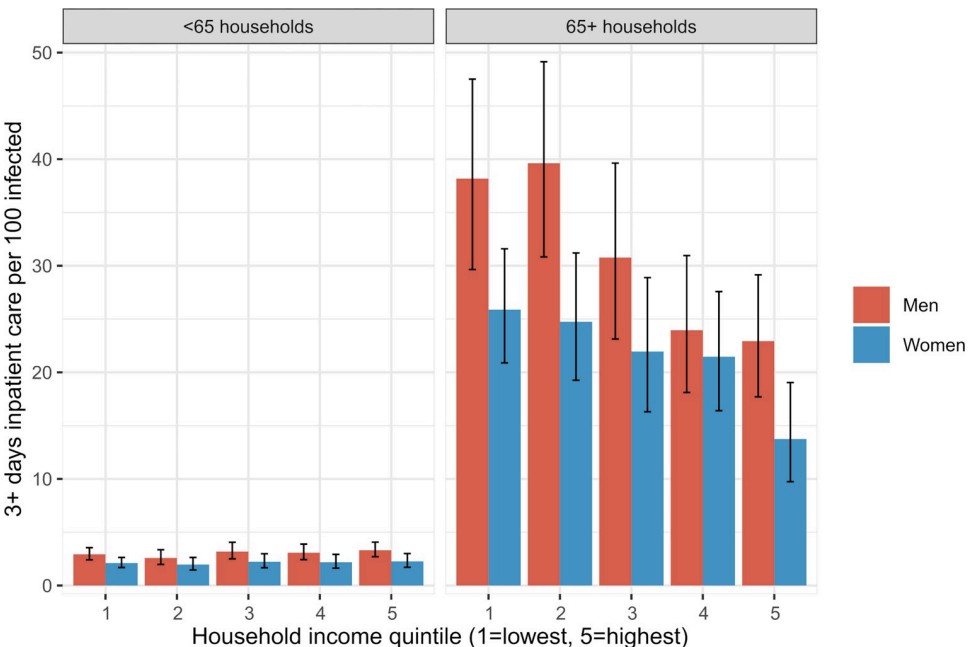

**Fig 3. Incidence of severe COVID-19 illness among those infected by sex and household income.** Incidence defined as 3+ days in hospital per 100 among those infected ($n = 24{,}138$) from 1 July to 31 December 2020 among individuals living in households with people aged under 65 and over 65. The whiskers represent 95% confidence intervals.

Fig 3 shows the risk of severe COVID-19 illness, defined as at least 3 consecutive days of inpatient care per 100 infected. The risk was small in the under-65 households (around 3%), and the income differences were modest. Among the over-65 households, on the other hand, the risks multiplied, and the income differences were larger. Men had a higher risk of severe illness than women in both age groups.

## Incidence models

When we controlled for age, sex, and regional characteristics in a 2-level regression model (Table 1, Model 1), there was an income gradient in COVID-19 incidence, the highest odds being among those in the lowest income quintile (OR 1.67, 95% CI 1.58 to 1.77, $p < 0.001$, 28.4% of infections) compared to those with the highest (18.9%). Neither a larger household size (Model 2) nor household-level work and school exposures (Model 3) attenuated the higher incidence among those with lower incomes. However, after including foreign background, the association of income and COVID-19 was largely attenuated (Model 4), with only the estimate of the lowest quintile (OR 1.23, 95% CI 1.16 to 1.30, $p < 0.001$) remaining statistically significant. The incidence of the lowest income quintile remained elevated in the fully adjusted Model 5 (OR 1.30, 95% CI 1.22 to 1.38, $p < 0.001$).

However, interaction between household income and foreign background status (Fig 4A) shows that this income gradient is only present among those with foreign background ($p$-value for interaction $<0.001$). Among individuals with a Finnish background, being in the lowest income quintile even appears as a moderate protective factor (OR 0.91, 95% CI 0.87 to 0.96, $p < 0.001$). People with foreign background in the lowest income households had a particularly high odds of COVID-19 infection (OR 3.81, 95% CI 3.61 to 4.02, $p < 0.001$ compared to people with a Finnish background in the highest income quintile). Adjustment for household

**Table 1. Odds ratios of COVID-19 infection from 1 July 2020 to 22 February 2021 among individuals living in under-65 households.**

| | Models | | | | |
|---|---|---|---|---|---|
| | **1*** | **2** | **3** | **4** | **5** |
| 1. Household income (ref. 5) (reference) | OR (95% CI) | OR (95% CI) | OR (95% CI) | OR (95% CI) | OR (95% CI) |
| Quintile 4 | 1.10 (1.03–1.16) | 1.10 (1.04–1.17) | 1.03 (0.97–1.10) | 1.04 (0.98–1.10) | 0.99 (0.93–1.05) |
| | $p = 0.002$ | $p = 0.001$ | $p = 0.28$ | $p = 0.22$ | $p = 0.65$ |
| Quintile 3 | 1.08 (1.01–1.14) | 1.07 (1.01–1.14) | 0.99 (0.94–1.06) | 0.97 (0.91–1.03) | 0.91 (0.85–0.97) |
| | $p = 0.02$ | $p = 0.03$ | $p = 0.87$ | $p = 0.35$ | $p = 0.003$ |
| Quintile 2 | 1.24 (1.17–1.32) | 1.23 (1.15–1.31) | 1.15 (1.08–1.22) | 1.05 (0.98–1.11) | 0.99 (0.93–1.06) |
| | $p < 0.001$ | $p < 0.001$ | $p < 0.001$ | $p = 0.15$ | $p = 0.82$ |
| Quintile 1 (lowest) | 1.67 (1.58–1.77) | 1.79 (1.69–1.91) | 1.69 (1.59–1.79) | 1.23 (1.16–1.30) | 1.30 (1.22–1.38) |
| | $p < 0.001$ | $p < 0.001$ | $p < 0.001$ | $p < 0.001$ | $p < 0.001$ |
| 2. Hospital district (ref. other) | | | | | |
| Helsinki Metropolitan (HUS) | 3.29 (3.17–3.43) | 3.43 (3.30–3.57) | 3.49 (3.35–3.63) | 2.91 (2.80–3.03) | 2.98 (2.86–3.10) |
| | $p < 0.001$ | $p < 0.001$ | $p < 0.001$ | $p < 0.001$ | $p < 0.001$ |
| 3. Urbanicity (ref. rural) | | | | | |
| Urban | 2.36 (2.23–2.49) | 2.49 (2.35–2.64) | 2.42 (2.28–2.57) | 2.11 (2.00–2.24) | 2.26 (2.13–2.39) |
| | $p < 0.001$ | $p < 0.001$ | $p < 0.001$ | $p < 0.001$ | $p < 0.001$ |
| Peri-urban | 1.09 (1.01–1.17) | 1.12 (1.05–1.21) | 1.12 (1.04–1.20) | 1.11 (1.03–1.19) | 1.13 (1.05–1.21) |
| | $p = 0.02$ | $p = 0.002$ | $p = 0.002$ | $p = 0.006$ | $p = 0.001$ |
| 4. Household size (ref. 1) | | | | | |
| 2 | 1.03 (0.98–1.09) | 1.17 (1.11–1.24) | | | 0.98 (0.93–1.04) |
| | $p = 0.21$ | $p < 0.001$ | | | $p = 0.48$ |
| 3 | 1.16 (1.09–1.23) | 1.33 (1.25–1.41) | | | 1.15 (1.08–1.23) |
| | $p < 0.001$ | $p < 0.001$ | | | $p < 0.001$ |
| 4 | 1.12 (1.06–1.19) | 1.33 (1.25–1.41) | | | 1.26 (1.16–1.36) |
| | $p < 0.001$ | $p < 0.001$ | | | $p < 0.001$ |
| 5+ | 1.84 (1.72–1.98) | 2.02 (1.88–2.18) | | | 1.92 (1.73–2.13) |
| | $p < 0.001$ | $p < 0.001$ | | | $p < 0.001$ |
| 5. Household-level work and school exposures | | | | | |
| (a) Lower nonmanual | 0.95 (0.92–0.99) | | 1.11 (1.06–1.15) | | 1.19 (1.14–1.24) |
| | $p = 0.007$ | | $p < 0.001$ | | $p < 0.001$ |
| (b) Self-employed | 1.05 (0.99–1.11) | | 1.15 (1.08–1.22) | | 1.08 (1.02–1.15) |
| | $p = 0.115$ | | $p < 0.001$ | | $p = 0.01$ |
| (c) Manual worker | 1.22 (1.17–1.27) | | 1.35 (1.29–1.41) | | 1.22 (1.17–1.28) |
| | $p < 0.001$ | | $p < 0.001$ | | $p < 0.001$ |
| (d) Student | 1.55 (1.49–1.61) | | 1.44 (1.39–1.50) | | 1.36 (1.30–1.42) |
| | $p < 0.001$ | | $p < 0.001$ | | $p < 0.001$ |
| (e) Child aged 13–15 | 1.33 (1.26–1.41) | | 1.30 (1.23–1.38) | | 1.03 (0.96–1.10) |
| | $p < 0.001$ | | $p < 0.001$ | | $p = 0.36$ |
| (f) Child aged 7–12 | 1.00 (0.95–1.05) | | 0.99 (0.94–1.04) | | 0.77 (0.72–0.82) |
| | $p = 0.93$ | | $p = 0.71$ | | $p < 0.001$ |
| (g) Child aged <7 | 0.88 (0.84–0.93) | | 0.92 (0.87–0.96) | | 0.68 (0.64–0.73) |
| | $p < 0.001$ | | $p = 0.001$ | | $p < 0.001$ |
| 6. Foreign background (ref. no) | | | | | |
| Yes | 4.75 (4.50–5.01) | | | 4.54 (4.30–4.80) | 4.59 (4.35–4.86) |
| | $p < 0.001$ | | | $p < 0.001$ | $p < 0.001$ |
| Household ICC | 0.770† | 0.772 | 0.771 | 0.768 | 0.771 |

(*Continued*)

**Table 1.** (Continued)

| | Models | | | | |
|---|---|---|---|---|---|
| | **1*** | **2** | **3** | **4** | **5** |
| | (0.766–0.774) | (0.768–0.776) | (0.767–0.774) | (0.765–0.772) | (0.767–0.775) |

*In Model 1, each variable adjusted separately for age and age squared, sex, and hospital district and urbanicity.

Model 2 adjusted for age and age squared, sex, hospital district and urbanicity, and household size.

Model 3 adjusted for age and age squared, sex, hospital district and urbanicity, household-level work, and school exposures.

Model 4 adjusted for age and age squared, sex, hospital district and urbanicity, and foreign background.

Model 5 adjusted for age and age squared, sex, hospital district and urbanicity, household size, household-level work and school exposures, and foreign background.

†Calculated from a model including age and age squared, sex, and hospital district and urbanicity.

CI, confidence interval; ICC, intraclass correlation; OR, odds ratio; p, p-value; Ref., reference category.

size and household-level work and school exposures did not substantially change the income gradient among those with foreign background (Fig 4B).

There was also a strong interaction between income and household size (Fig 5A), with low income being a much stronger risk factor in large households (p-value for interaction <0.001). Households in the lowest income quintile with 5 or more members stood out with a particularly high odds of infection (OR 3.72, 95% CI 3.35 to 4.12, p < 0.001 compared with single-person households in the highest income quintile). Following adjustment for household-level work and school exposures and foreign background (Fig 5B), the income gradient was considerably attenuated across all household sizes but the incidence remained high especially in the poorest households with 5 or more members (p-value for interaction <0.001). An additional

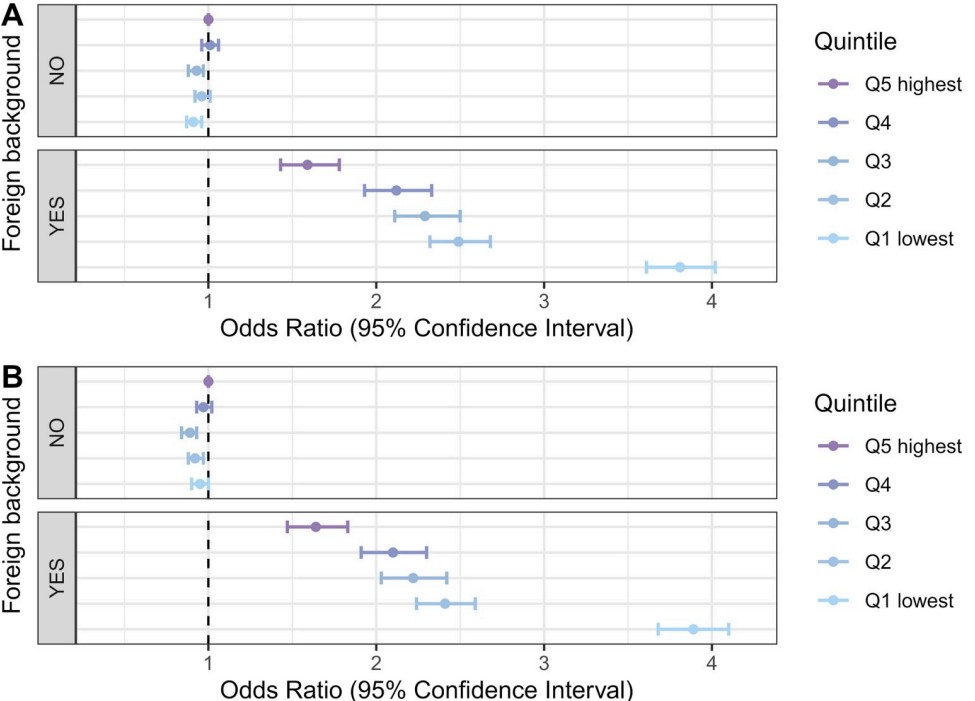

**Fig 4. Odds ratios of COVID-19 incidence by household income quintile and foreign background status.** In under-65 households (A) adjusted for age and age squared, sex, and regional characteristics; (B) adjusted for age and age squared, sex, regional characteristics, household size, and household-level work and school exposures.

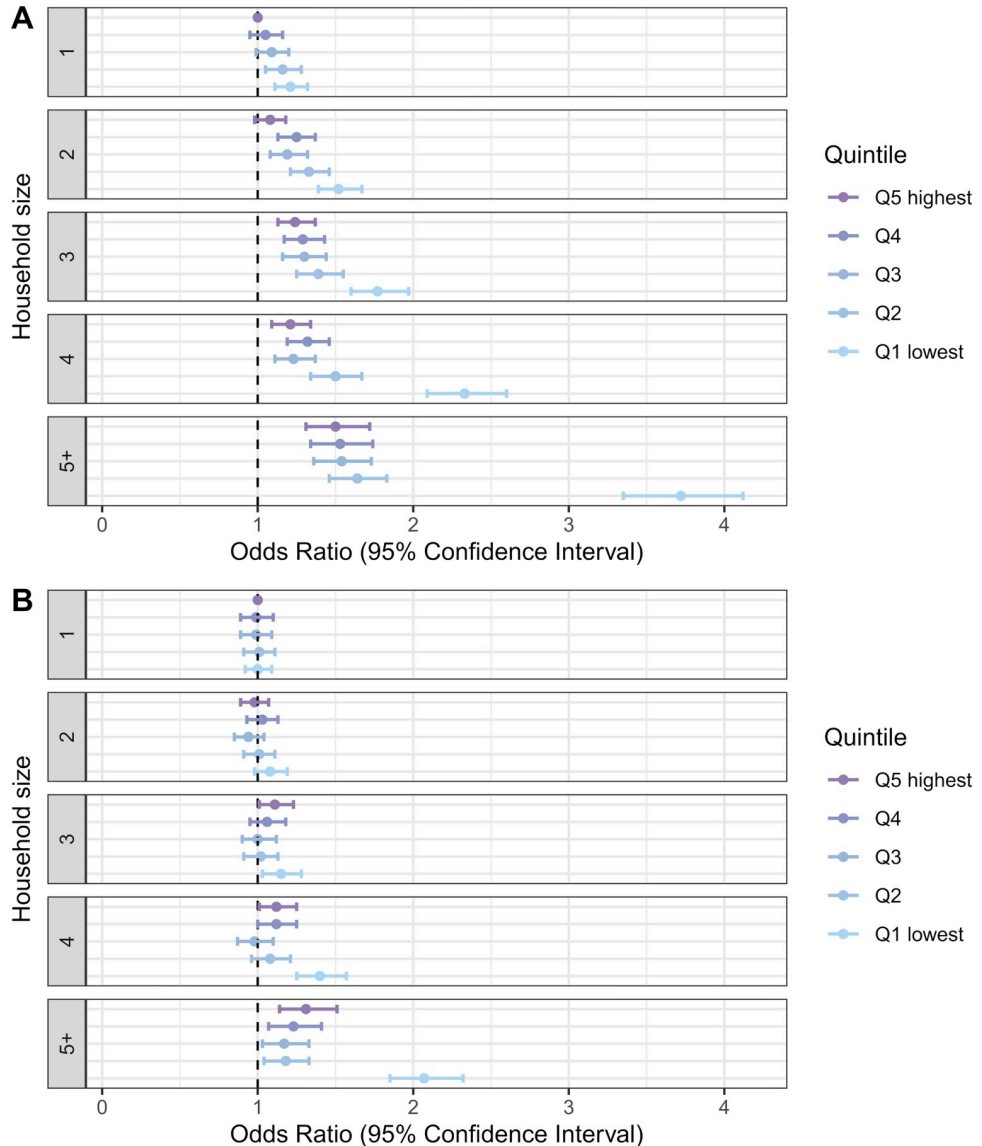

**Fig 5. Odds ratios of COVID-19 incidence by household income quintile and household size.** In under-65 households (A) adjusted for age and age squared, sex, and regional characteristics; (B) adjusted for age and age squared, sex, regional characteristics, household-level work and school exposures, and foreign background.

interaction analysis between household income, household size, and foreign background (S1 Fig) indicated that the excess risk of large poor household was present only among those with foreign background.

The high ICC (about 0.77 in all the models) indicates that if 1 household member was infected, the others were very likely to become infected as well.

## Severity models

When we controlled for age, sex, and regional characteristics (Table 2, Model 1), the odds of severe COVID-19 illness was twice as high in the lowest household income quintile (OR 1.97, 95% CI 1.52 to 2.56, $p < 0.001$, 28.0% of infections) compared to the highest (21.6%).

**Table 2. Odds ratios of severe COVID-19 illness among those infected from 1 July to 31 December 2020, individuals living in under-65 households.**

| | Models | | | | | |
|---|---|---|---|---|---|---|
| | 1* | 2 | 3 | 4 | 5 | 6 |
| 1. Household income (ref. 5) (reference) | OR (95% CI) | OR (95% CI) | OR (95% CI) | OR (95% CI) | OR (95% CI) | OR (95% CI) |
| Quintile 4 | 1.05 (0.79–1.38) | 1.03 (0.78–1.36) | 1.03 (0.78–1.37) | 0.96 (0.72–1.28) | 0.99 (0.75–1.31) | 0.94 (0.71–1.26) |
| | $p = 0.74$ | $p = 0.86$ | $p = 0.82$ | $p = 0.77$ | $p = 0.94$ | $p = 0.69$ |
| Quintile 3 | 1.25 (0.94–1.65) | 1.19 (0.90–1.59) | 1.20 (0.90–1.60) | 1.07 (0.79–1.44) | 1.13 (0.84–1.50) | 1.03 (0.76–1.39) |
| | $p = 0.12$ | $p = 0.23$ | $p = 0.22$ | $p = 0.67$ | $p = 0.42$ | $p = 0.87$ |
| Quintile 2 | 1.23 (0.92–1.65) | 1.14 (0.84–1.53) | 1.14 (0.84–1.53) | 0.96 (0.70–1.31) | 1.02 (0.75–1.38) | 0.88 (0.64–1.21) |
| | $p = 0.16$ | $p = 0.40$ | $p = 0.41$ | $p = 0.80$ | $p = 0.91$ | $p = 0.43$ |
| Quintile 1 (lowest) | 1.97 (1.52–2.56) | 1.84 (1.41–2.40) | 1.80 (1.37–2.36) | 1.37 (1.01–1.86) | 1.45 (1.08–1.94) | 1.08 (0.77–1.52) |
| | $p < 0.001$ | $p < 0.001$ | $p < 0.001$ | $p = 0.04$ | $p = 0.01$ | $p = 0.64$ |
| 2. Hospital district (ref. other) | | | | | | |
| Helsinki Metropolitan (HUS) | 0.76 (0.63–0.92) | 0.77 (0.64–0.93) | 0.77 (0.64–0.94) | 0.78 (0.64–0.94) | 0.74 (0.61–0.89) | 0.75 (0.61–0.91) |
| | $p = 0.004$ | $p = 0.007$ | $p = 0.009$ | $p = 0.01$ | $p = 0.002$ | $p = 0.003$ |
| 3. Urbanicity (ref. rural) | | | | | | |
| Urban | 1.01 (0.76–1.33) | 0.93 (0.70–1.24) | 0.93 (0.70–1.23) | 0.94 (0.70–1.25) | 0.88 (0.66–1.17) | 0.87 (0.65–1.17) |
| | $p = 0.97$ | $p = 0.62$ | $p = 0.61$ | $p = 0.66$ | $p = 0.38$ | $p = 0.37$ |
| Peri-urban | 0.73 (0.50–1.05) | 0.73 (0.50–1.07) | 0.73 (0.50–1.07) | 0.74 (0.50–1.08) | 0.73 (0.50–1.07) | 0.74 (0.51–1.09) |
| | $p = 0.09$ | $p = 0.10$ | $p = 0.11$ | $p = 0.12$ | $p = 0.11$ | $p = 0.13$ |
| 4. Comorbidities | | | | | | |
| (a) Cancer | 1.65 (0.93–2.94) | 1.48 (0.81–2.71) | 1.49 (0.81–2.72) | 1.49 (0.81–2.75) | 1.52 (0.83–2.77) | 1.51 (0.82–2.79) |
| | $p = 0.09$ | $p = 0.20$ | $p = 0.20$ | $p = 0.20$ | $p = 0.18$ | $p = 0.18$ |
| (b) Kidney failure | 19.46 (6.04–62.70) | 10.58 (3.09–36.14) | 10.33 (3.02–35.33) | 9.09 (2.63–31.46) | 10.56 (3.09–36.11) | 8.78 (2.53–30.41) |
| | $p < 0.001$ | $p < 0.001$ | $p < 0.001$ | $p < 0.001$ | $p < 0.001$ | $p = 0.001$ |
| (c) Chronic lung disease | 2.37 (1.74–3.22) | 2.35 (1.71–3.22) | 2.34 (1.70–3.21) | 2.30 (1.67–3.17) | 2.46 (1.79–3.38) | 2.39 (1.73–3.30) |
| | $p < 0.001$ | $p < 0.001$ | $p < 0.001$ | $p < 0.001$ | $p < 0.001$ | $p < 0.001$ |
| (d) Diabetes | 2.29 (1.73–3.03) | 1.86 (1.39–2.49) | 1.87 (1.39–2.50) | 1.82 (1.35–2.44) | 1.83 (1.36–2.45) | 1.80 (1.34–2.41) |
| | $p < 0.001$ | $p < 0.001$ | $p < 0.001$ | $p < 0.001$ | $p < 0.001$ | $p < 0.001$ |
| (e) Chronic heart disease | 1.82 (1.33–2.49) | 1.35 (0.96–1.88) | 1.35 (0.96–1.88) | 1.31 (0.93–1.85) | 1.41 (1.01–1.97) | 1.37 (0.97–1.93) |
| | $p < 0.001$ | $p = 0.08$ | $p = 0.08$ | $p = 0.12$ | $p = 0.05$ | $p = 0.07$ |
| (f) Psychotic disorders | 2.08 (1.07–4.02) | 1.58 (0.80–3.13) | 1.56 (0.78–3.09) | 1.14 (0.56–2.30) | 1.77 (0.89–3.51) | 1.21 (0.60–2.46) |
| | $p = 0.03$ | $p = 0.19$ | $p = 0.21$ | $p = 0.72$ | $p = 0.10$ | $p = 0.59$ |
| 5. Household size (ref. 1) | | | | | | |
| 2 | 0.87 (0.68–1.12) | | 0.97 (0.75–1.26) | | | 0.88 (0.68–1.15) |
| | $p = 0.28$ | | $p = 0.84$ | | | $p = 0.35$ |
| 3 | 0.76 (0.57–1.02) | | 0.85 (0.63–1.14) | | | 0.77 (0.57–1.04) |
| | $p = 0.07$ | | $p = 0.28$ | | | $p = 0.08$ |
| 4 | 0.72 (0.53–0.98) | | 0.80 (0.59–1.09) | | | 0.73 (0.53–1.00) |
| | $p = 0.04$ | | $p = 0.16$ | | | $p = 0.05$ |
| 5+ | 1.04 (0.77–1.40) | | 1.01 (0.74–1.37) | | | 0.87 (0.63–1.19) |
| | $p = 0.80$ | | $p = 0.96$ | | | $p = 0.37$ |
| 6. Occupation (ref. upper nonmanual) | | | | | | |
| Lower nonmanual | 1.13 (0.84–1.54) | | | 1.07 (0.78–1.46) | | 1.05 (0.76–1.44) |
| | $p = 0.42$ | | | $p = 0.68$ | | $p = 0.77$ |

(Continued)

**Table 2.** (Continued)

| | Models | | | | | |
|---|---|---|---|---|---|---|
| | 1* | 2 | 3 | 4 | 5 | 6 |
| Self-employed | 1.22 (0.82–1.82) | | | 1.15 (0.76–1.73) | | 1.12 (0.74–1.69) |
| | $p = 0.33$ | | | $p = 0.51$ | | $p = 0.60$ |
| Manual worker | 1.26 (0.92–1.72) | | | 1.19 (0.86–1.66) | | 1.08 (0.77–1.51) |
| | $p = 0.15$ | | | $p = 0.29$ | | $p = 0.65$ |
| Student | 1.22 (0.76–1.98) | | | 1.03 (0.62–1.71) | | 0.94 (0.56–1.56) |
| | $p = 0.41$ | | | $p = 0.91$ | | $p = 0.81$ |
| Pensioner | 3.67 (2.48–5.43) | | | 2.77 (1.81–4.25) | | 2.70 (1.76–4.15) |
| | $p < 0.001$ | | | $p < 0.001$ | | $p < 0.001$ |
| Other/unknown | 2.04 (1.50–2.77) | | | 1.62 (1.14–2.30) | | 1.44 (1.01–2.06) |
| | $p < 0.001$ | | | $p = 0.007$ | | $p = 0.05$ |
| 7. Foreign background (ref. no) | | | | | | |
| Yes | 1.74 (1.43–2.12) | | | | 1.55 (1.23–1.96) | 1.62 (1.26–2.06) |
| | $p < 0.001$ | | | | $p < 0.001$ | $p < 0.001$ |
| Household ICC | 0.185† | 0.202 | 0.204 | 0.216 | 0.202 | 0.214 |
| | (0.075–0.392) | (0.088–0.399) | (0.090–0.398) | (0.101–0.403) | (0.088–0.400) | (0.099–0.401) |

*In Model 1, each variable adjusted separately for age and age squared, sex, and hospital district and urbanicity.

Model 2 adjusted for age and age squared, sex, hospital district and urbanicity, and comorbidities.

Model 3 adjusted for age and age squared, sex, hospital district and urbanicity, comorbidities, and household size.

Model 4 adjusted for age and age squared, sex, hospital district and urbanicity, comorbidities, and personal occupation.

Model 5 adjusted for age and age squared, sex, hospital district and urbanicity, comorbidities, and foreign background.

Model 6 adjusted for age and age squared, sex, hospital district and urbanicity, comorbidities, household size, personal occupation, and foreign background.

†Calculated from a model including age and age squared, sex, and hospital district and urbanicity.

CI, confidence interval; ICC, intraclass correlation; OR, odds ratio; p, p-value; Ref., reference category.

These income differences were not attributable to comorbidities (Model 2) or household size (Model 3). They were, however, attributable in part to individual-level occupational status (Model 4) and foreign background (Model 5). In the fully adjusted Model 6, the other risk factors attenuated all the income differences in the risk of severe illness.

There were too few observations to draw reliable conclusions about the interaction of household income with foreign background (Fig 6) or household size (Fig 7). However, these effects appear weak and inconsistent. The ICC (about 0.20) indicated strong household clustering of severe COVID-19, even when we controlled for the individual-level and household-level risk factors.

## Sensitivity analyses

In case severity models, defining the outcome as any inpatient care instead of a care episode of at least 3 days increased the number of severe cases by about 10%, but had little impact on the results (S2 Table). Results from the models where the outcome was defined to include only primary COVID-19 diagnoses were also highly similar to our main analyses including both primary and secondary diagnoses (S3 Table).

Results from severity models conducted among the full population instead of those infected reflect income differences in both incidence and case severity (S4 Table), and the estimates lay in between the corresponding estimates from our main results on incidence and case severity.

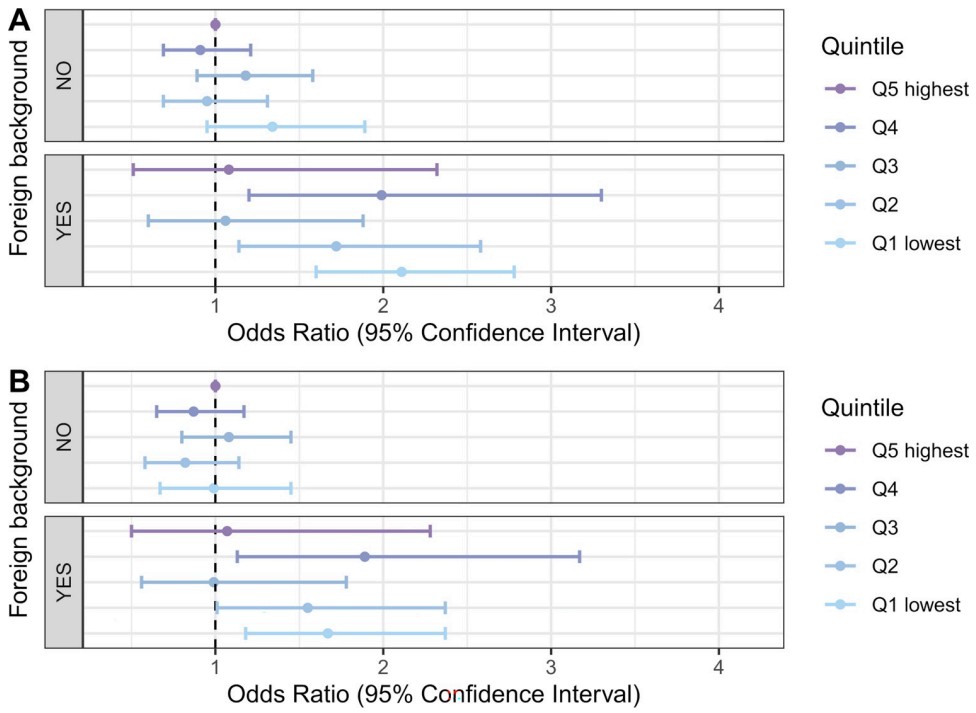

**Fig 6. Odds ratios of severe COVID-19 illness among those infected by household income quintile and foreign background status.** In under-65 households (A) adjusted for age and age squared, sex, regional characteristics, and comorbidities; (B) adjusted for age and age squared, sex, regional characteristics, comorbidities, household size, and household-level work and school exposures.

## Discussion

We used total population data covering over 4 million individuals nested within households to assess the associations of household income with COVID-19 incidence and severity, and to quantify the clustering of COVID-19 in working-age households. In line with prior evidence, we found that individuals living in low-income households had higher risk of both COVID-19 incidence and severity—however, this was largely driven by the foreign background of household members. In fact, a separate analysis revealed a strong income gradient among individuals with foreign background only, and no income association at all among those with native background. The odds of severe illness among the infected were likewise highest among those with lowest income, but this association was also strongly driven by other risk factors: comorbidities, personal occupational status, and foreign background. Both incidence and severity were strongly clustered in households: Around 77% of the variation in incidence and 20% of the variation in severity were attributable to differences between households.

### Comparison with other studies

To our knowledge, this is the first study to assess income and other sociodemographic risk factors for COVID-19 incidence and severity using both household and individual-level data. Incorporating the household level is a major contribution, because, as we show, not only infections, but also severe illness is strongly clustered within households.

Our general finding of a higher COVID-19 incidence in low-income households is in accordance with previous results associating incidence with area-level measures of low education, deprivation [1,6], and low mean income by district [2,3]. Our study differed from many

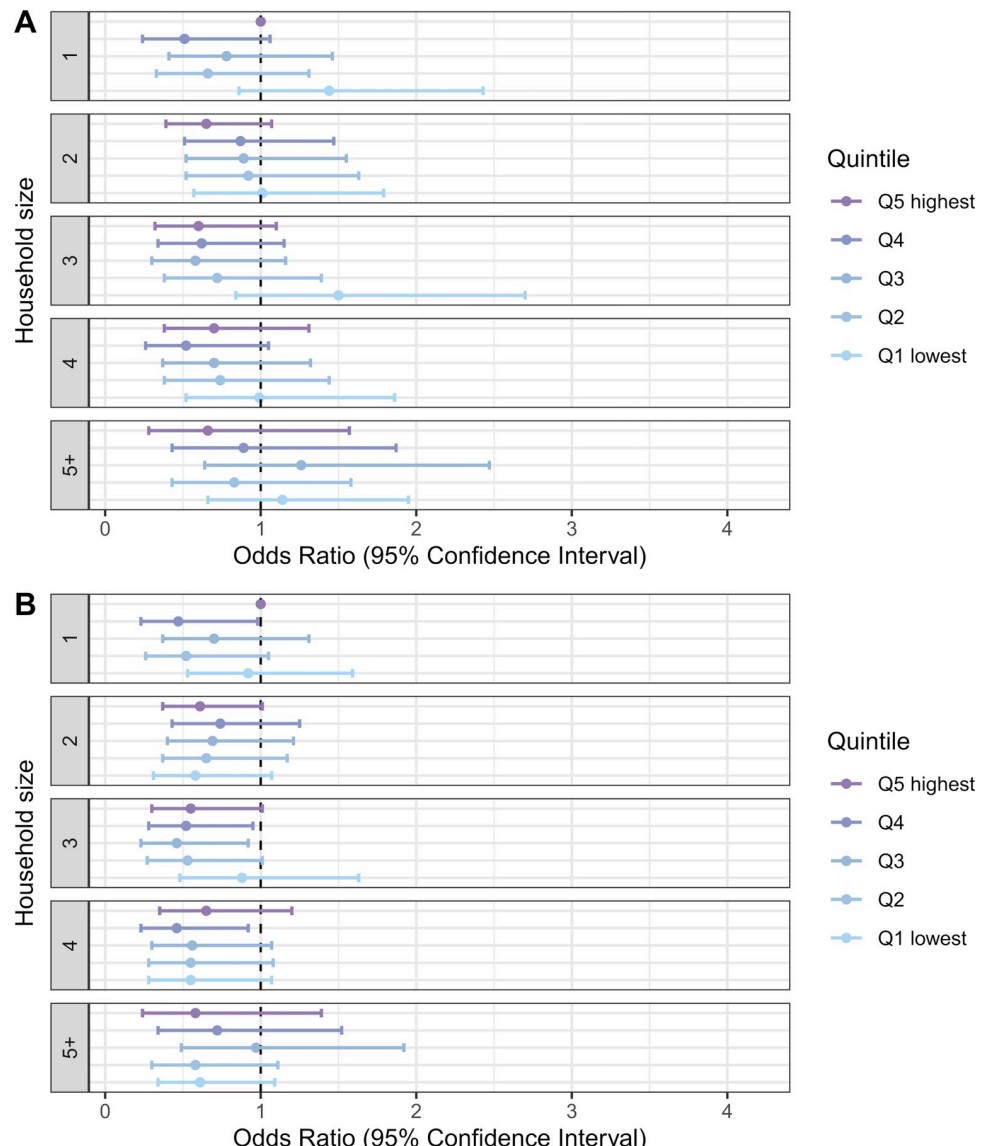

**Fig 7. Odds ratios of severe COVID-19 illness among those infected by household income quintile and household size.** In under-65 households (A) adjusted for age and age squared, sex, regional characteristics, and comorbidities; (B) adjusted for age and age squared, sex, regional characteristics, comorbidities, household-level work and school exposures, and foreign background.

others in that we were able to control for a set of other important individual- and household-level risk factors, and indeed we found that having foreign background was a major driver of the income differences, while other risk factors for higher COVID-19 incidence made no difference for the income association. Furthermore, we showed that the income gradient was only present among people with foreign background and nonexistent among those with a Finnish background. New to the existing literature, this finding has no direct point of reference from previous studies. While foreign background and ethnicity have been linked to higher COVID-19 incidence and severity in many previous studies [18], to our knowledge, only 1 prior study has assessed the role of foreign background or ethnicity in the association between income and COVID-19 outcomes [1]. Their results contrast with ours: In this study based on

UK Biobank data, controlling for ethnicity and country of birth explained little of the higher incidence in socioeconomically deprived areas [1]. The differing results may relate to the selective sample and area-based measurement of socioeconomic exposures in the UK Biobank data, compared with our total population data with household-based measures.

Our results suggest that, overall, there is no independent association between household income and COVID-19 incidence. Low income was related to higher COVID-19 incidence only among people with foreign background, and this association was independent of household size and the work and school exposures that we measured. The combination of foreign background and low income is likely to capture vulnerabilities related to race, ethnic minority, and refugee background which may influence infection risk through material, social, and behavioural mechanisms. Prior studies from the United States indicate that people with low income have fewer material and social resources to protect themselves from COVID-19 infection [27]. Social mechanisms may relate to lower health literacy, language barriers, racism, and structural discrimination faced by people with foreign background and low socioeconomic position [28–30]. Higher susceptibility to COVID-19 may also relate to a lowered immune response due to higher stress levels [14,31]. Finally, behavioural factors such as possible broader social interactions among people with foreign background [29] may further add to the clustering of risk factors among those with foreign background and low income. Due to the epidemic nature of COVID-19, the extent of social interactions could be of particular importance if outbreaks are concentrated in schools and neighborhoods having more people with low socioeconomic position and foreign background. The lack of neighborhood-level controls is a limitation of our study.

A major contribution of our study is that we were able to assess the risk factors for case severity in the total infected population. Previous studies on severe COVID-19 have either limited the study population to those already hospitalized or seeking health care [10,11,19], or analysed the general population irrespective of infection status [7–9]. The former target a group that is already selected on case severity whereas the latter conflate the risk factors of incidence and severity. A study on the general population of Sweden, for example, found that low income and educational level predicted increased COVID-19 mortality [7], but these estimates may reflect socioeconomic differences in either incidence or case fatality, or both. Moreover, few previous studies have been able to control for important confounders. A Swiss study reported higher case severity and fatality rates among the infected in neighbourhoods with a low socioeconomic index based on 2,000 census data, but it did not adjust for individual risk factors other than age and sex [12]. In contrast, our adjusted results indicate that household income is not independently associated with case severity. In fact, the higher risk of severe COVID-19 in low-income households was strongly driven by personal occupational status and foreign background. Personal occupational status is likely to capture health-related confounding by controlling for being on early-age pension or having an unknown personal occupational status. Foreign background, in turn, may capture ethnic differences in the risk for severe COVID-19, which could relate to a complex set of factors such as racism and structural discrimination, or barriers in access to care that we were not able to measure [18,32]. Another reason for our results may relate to selection bias caused by differential testing [33]. If people with foreign background tend to test less for mild COVID-19 symptoms, this will lead to a disproportional share of severe cases among those identified as infected in this group. Further studies should investigate the mechanisms producing social and ethnic differences in case severity, preferably in samples where infections are identified by screening rather than by self-selective testing.

We found very high infection clustering within households, corroborating previous evidence that the risk of COVID-19 infection is higher within households than in other social

contexts [21,22]. Likewise, our finding that large low-income households had the highest incidence implies an accumulation of risk factors in specific types of household, and supports the earlier observation on the disadvantages of household crowding [5]. Our results further suggest that this accumulation mainly occurs among the population with foreign background. Our study is the first to assess the household clustering of case severity. The correlation in the likelihood of severe illness between household members was around 20%, which the measured individual-level risk factors failed to explain. However, unmeasured risk factors such as general health, obesity, smoking, and other health behaviours may cluster in households and offer some explanation of why several members of some households tend to have severe COVID-19. Shared genetic vulnerability is unlikely to explain much of the household clustering of severity because the vast majority of multigenerational households in our data consist of parents and their underage children, and severe COVID-19 is rare among the young.

### The strengths and weaknesses of the study

The unique strength of our study lies in the use of total population register data comprising individuals nested within households. This enabled us to assess household-level risk factors for COVID-19 while properly taking into account the household clustering of outcomes [24]. Furthermore, we were able to quantify the household clustering for both incidence and case severity. The use of up-to-date administrative data provided us with a more accurate measurement of the socioeconomic and other risk factors at both individual and household level. This is an important contribution as the existing evidence is mostly based on area-level socioeconomic measures [1–3,5,6,8–13,19], or rely on measures from more than a decade ago [34,12]. Of the common indicators of socioeconomic position (education, income, deprivation), we chose to focus on household income. It is reliably measured and available for all individuals irrespective of their age, employment or immigrant status, and well-suited for household-level analyses. We also had access to data on annual household income from a time point close to the pandemic, 2018. However, a limitation of our study is that household income captures only one aspect of the multidimensional concept of socioeconomic position. While our study also included information on occupational status, future research incorporating multiple dimensions of socioeconomic position is needed for a more comprehensive picture of the social inequalities in COVID-19 outcomes.

Consistent information on laboratory-confirmed infections and hospital care records allowed us to study both incidence and case severity. Finland's testing strategy during the study period was to include even the mildly symptomatic. The tests were free of charge and widely available, although waiting times still varied during the late summer of 2020 [35]. It has been estimated that laboratory-confirmed COVID-19 cases in early 2021 represented at least half of the total infections in Finland [36]. Such an underestimation of cases could lead to bias if there were differences in the testing threshold according to our key variables of interest. We have no direct way of assessing the magnitude of this bias. However, evidence from Switzerland [12] indicates that test uptake may be lower among more disadvantaged population subgroups and may thus lead to the underestimation of income differentials in incidence. Furthermore, if people with a higher income were tested for milder symptoms, this may lead to the overestimation of income differentials in illness severity among the infected [37]. It would be valuable in future research to obtain direct evidence of testing frequency and the true infection prevalence in specific subpopulations.

Our data cover a period before vaccinations were available to the under 65. Although the social dynamics of COVID-19 infections and outcomes will most likely change as the rates of vaccination increase and new variants of the virus emerge, the significance of the household

context of individuals is unlikely to diminish. Moreover, the clustering of severe illness could have long-lasting effects in the households as the severe illness increases the risk for long COVID symptoms [38]. The potential impact of socioeconomic differentials in vaccination take up on COVID-19 incidence and severity will also be a relevant topic for future studies.

## Conclusions

We showed that people with a low household income are at higher risk of COVID-19 incidence and case severity. However, these income differences in incidence were only present among the population with foreign background, while there was no association between income and COVID-19 incidence among those with native background. The income differences found in case severity were also strongly driven by other individual and household-level risk factors, foreign background in particular. Socioeconomic position as reflected by household income may thus not be an independent risk factor for COVID-19 outcomes among people with a native background. However, people with foreign background living in low-income households emerged as a particularly vulnerable group to consider when planning preventive measures. Both incidence and case severity are strongly clustered within households. This highlights the importance of the household context in understanding the microlevel dynamics of the COVID-19 pandemic.

## Supporting information

**S1 Checklist. STROBE checklist.**
(DOCX)

**S1 Table. The distributions of the population at risk and cases in the incidence and severity analyses by risk factors.**
(DOCX)

**S2 Table. Odds ratios of hospitalization with COVID-19 diagnosis ($N$ = 696) among those infected from 1 July to 31 December 2020 ($N$ = 24,138), individuals living in under-65 households.** Hospitalization is defined as any admission to the hospital with a COVID-19 diagnosis. Results are from 2-level logistic regressions, with individuals at level 1 nested in households at level 2. All models are adjusted for age and age squared and sex.
(DOCX)

**S3 Table. Odds ratios of severe illness with COVID-19 as the primary diagnosis ($N$ = 387) among those infected from 1 July to 31 December 2020 ($N$ = 24,138), individuals living in under-65 households.** Severe illness is defined as having at least 3 consecutive days of inpatient care with a COVID-19 diagnosis. Results are from 2-level logistic regressions, with individuals at level 1 nested in households at level 2. All models are adjusted for age and age squared and sex.
(DOCX)

**S4 Table. Odds ratios of severe COVID-19 illness ($N$ = 636) from 1 July to 31 December 2020, among all individuals living in under-65 households ($N$ = 4,315,342).** Severe illness is defined as having at least 3 consecutive days of inpatient care with a COVID-19 diagnosis. Results are from 1-level logistic regressions with household-clustered standard errors. All models are adjusted for age and age squared and sex.
(DOCX)

**S1 Fig. Odds ratios of COVID-19 incidence by household income quintile, household size, and foreign background status in under-65 households.** Adjusted for age and age squared,

sex, regional characteristics, and household-level work and school exposures.
(TIF)

## Author Contributions

**Conceptualization:** Sanni Saarinen, Heta Moustgaard, Hanna Remes, Riikka Sallinen, Pekka Martikainen.

**Data curation:** Sanni Saarinen, Riikka Sallinen.

**Formal analysis:** Sanni Saarinen, Heta Moustgaard, Hanna Remes, Pekka Martikainen.

**Funding acquisition:** Pekka Martikainen.

**Investigation:** Sanni Saarinen, Heta Moustgaard, Hanna Remes, Pekka Martikainen.

**Methodology:** Sanni Saarinen, Heta Moustgaard, Hanna Remes, Riikka Sallinen, Pekka Martikainen.

**Project administration:** Pekka Martikainen.

**Resources:** Riikka Sallinen, Pekka Martikainen.

**Supervision:** Heta Moustgaard, Hanna Remes, Pekka Martikainen.

**Validation:** Sanni Saarinen, Heta Moustgaard, Hanna Remes, Pekka Martikainen.

**Visualization:** Sanni Saarinen.

**Writing – original draft:** Sanni Saarinen, Heta Moustgaard, Hanna Remes, Riikka Sallinen, Pekka Martikainen.

**Writing – review & editing:** Sanni Saarinen, Heta Moustgaard, Hanna Remes, Riikka Sallinen, Pekka Martikainen.

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
