## [Editor Report · Decision Letter 0]

8 Oct 2021

Dear Dr Saarinen, 

Thank you for submitting your manuscript entitled "Income differences in COVID-19 incidence and severity: a nationwide register study of over four million individuals nested in households" for consideration by PLOS Medicine.

Your manuscript has now been evaluated by the PLOS Medicine editorial staff and I am writing to let you know that we would like to send your submission out for external peer review.

Please include line numbers in your manuscript and re-submit it within two working days, i.e. by Oct 12 2021 11:59PM.

Kind regards,

Beryne Odeny

PLOS Medicine

---

## [Decision Letter · Decision Letter 1]

17 Feb 2022

Dear Dr. Saarinen,

Thank you very much for submitting your manuscript "Income differences in COVID-19 incidence and severity: a nationwide register study of over four million individuals nested in households" (PMEDICINE-D-21-04207R1) for consideration at PLOS Medicine. 

Your paper was evaluated by a senior editor and discussed among all the editors here. It was also sent to independent reviewers, including a statistical reviewer. The reviews are appended at the bottom of this email and any accompanying reviewer attachments can be seen via the link below:

[LINK]

In light of these reviews, I am afraid that we will not be able to accept the manuscript for publication in the journal in its current form. We would be grateful if you could please revise your manuscript to respond to comments raised by reviewers. We would strongly recommend that you pay special attention to reviewer #3 comments regarding the conceptualization and framing of your research question. Please note that this is not a guarantee that we will accept the manuscript and that further consideration is dependent on the submission of a manuscript that addresses all reviewer concerns. We will carefully review your manuscript upon revision, so please ensure that your revision is thorough.

We expect to receive your revised manuscript by Mar 09 2022 11:59PM. Please email us (plosmedicine@plos.org) if you have any questions or concerns.

We look forward to receiving your revised manuscript. 

Sincerely,

Beryne Odeny, 

PLOS Medicine

plosmedicine.org

1) Please revise your title according to PLOS Medicine's style. Your title must be nondeclarative and not a question. It should begin with main concept if possible. Please place the study design in the subtitle (i.e., after a colon), e.g., cross-sectional population-based study. Include country in the title. For example, “ Income differences, COVID-19 incidence and severity in Finland: A cross-sectional population-based study”

2) The Data Availability Statement (DAS) requires revision. For each data source used in your study: 

a) If the data are freely or publicly available, state the location of the data: within the paper, in Supporting Information files, or in a public repository (include the DOI or accession number).

3) Please strongly consider reviewer #3’s comments on your conceptualization of the research question and analyses. The use of DAGs and mediation analyses to untangle causal associations will be critical to refining and strengthening your findings and conclusions. Of note, “foreign background” is not unrelated to ethnicity and this has been explored in other studies.

4) Abstract:

a) Please combine the Methods and Findings sections into one section, “Methods and Findings”

b) Please ensure that all numbers presented in the abstract are present and identical to numbers presented in the main manuscript text.

c) Please include the actual amounts and/or absolute risk(s) of relevant outcomes, not just coefficients.

d) Please quantify the main results (please present both 95% CIs and p values).

e) In the last sentence of the Abstract Methods and Findings section, please describe the main limitation(s) of the study's methodology.

6) Please avoid assertions of primacy (“This is the first study to ..."). Instead, state “to our knowledge” or similar.

7) Please remove the last sentence of your introduction (“We adjust our analyses …”) and move it the statistical analysis section 

8) Did your study have a prospective protocol or analysis plan? Please state this (either way) early in the Methods section. 

9) Please ensure that the study is reported according to the STROBE guideline, and include the completed STROBE checklist as Supporting Information. Please add the following statement, or similar, to the Methods: "This study is reported as per the Strengthening the Reporting of Observational Studies in Epidemiology (STROBE) guideline (S1 Checklist)."

10) How was race/ethnicity defined and by whom? 

11) Please provide p values in addition to 95% CIs in the main text and tables

12) Throughout the text, please remove language that implies causality, such as “impact” , “predict” in the discussion and conclusion sections. Refer to associations instead.

13) Please provide the meaning of the bars and whisker in your figures.

14) References: 

a) Please select the PLOS Medicine reference style in your citation manager. In-text reference call outs should be presented as follows noting the absence of spaces within the square brackets, e.g., "... services [1,2]."

b) References should have no more than six names. For those with more than six names, please ensure that et al., is inserted after six names

Comments from the reviewers:

Reviewer #1: "Income differences in COVID-19 incidence and severity: a nationwide register study of over four million individuals nested in households" analyzes Finnish national data, primarily towards identifying the effect of income differences on COVID-19 incidence and severity. The main conclusions were that low household income was a strong risk factor for both COVID-19 incidence and case severity, which is further highly attributable to a foreign background (Line 252). Particular strengths of the study include the availability of total population register data at the household level, and fairly complete data on common risk factors. The findings generally collaborate previous work on the impact of low household income on COVID-19 infection risk. Nevertheless, a number of points might be clarified:

1. In Line 96, it is stated that the main analyses are on COVID-19 infections (amongst the full population), and risk of severe illness (only amongst the infected). It might be clarified further as to whether the risk of severe illness amongst the full population would be a relevant analysis, and if so, it might be considered to be undertaken.

2. In Line 128, it is stated that severe illness was defined as inpatient care lasting at least three consecutive days with a primary or secondary diagnosis of COVID-19. Sensitivity analysis (with accompanying demographics presented) for primary vs. secondary diagnosis might be considered, if possible.

3. In Line 135, it is stated that household size was categorized as 1, 2, 3 or 4+. From Supplementary Table 1, 4+ household size accounts for 36.8% of the population at risk. Sensitivity analysis at higher granularities (i.e. 5+, 8+, etc.) might be considered if the data is available.

4. In Line 181, it is stated that two-level logistic regression models (individuals nested in households) was used. A brief explanation of the construction/implementation of such two-level models might be provided, possibly in supplementary material.

5. In Line 186, it is stated that for incidence, Models 2 to 5 were built on Model 1 (Table 1). However, Table 1 appears to show missing values for entire groups for Models 2 to 4 in particular (suggesting that they may not be entirely "built on" Model 1?), and it is not immediately clear as to exactly what was included/adjusted for, in each of these models (as also relating to the previous point). As such, the full definition of the various models in Tables 1 & 2 - including any interaction variables - might be provided in supplementary material (and perhaps briefly clarified in the main text), if possible.

6. The choice of which categories/groups adjusted for in each model (as in Tables 1 & 2) might be explained further, if possible. For example, while foreign background was adjusted for and found to account for much of the effect from income differences, it appears possible that other risk factors such as urbanicity (urban with consistently significantly higher risk, may also be expected to correlate highly with foreign background) might have similar effects.

7. In Line 317, the limitation of laboratory-confirmed COVID-19 cases likely not representing the full incidence, and potential bias in the distribution of laboratory-confirmed cases, was acknowledged. Related to this, the expected coverage of the national register used for incidence/severe illness might be briefly described. In particular, does the register cover all relevant medical facilities, or does it possibly exclude some categories, e.g. private clinics?

8. For Tables 1 & 2, the commonly-recognized major risk factors of age and sex appear not to have their odds ratios included. This might be addressed if possible (with accompanying demographics included in Supplementary Table 1)

9. For Tables 1 & 2, the top row might be captioned as being the Model(s), for greater clarity.

10. From Figure 2, COVID-19 incidence is significantly lower in 65+ households, as compared to <65 households. This observation might be briefly commented on, if possible.

Reviewer #2: This is an interesting and important study.

I do have the following question and comment:

Why the time range of the two cohorts is different?

The length of hospital stay is not a criterion for COVID-19 severity. There are other clinical criteria the most important of which is saturation level. With regard to this I think it is better to analyze incidence of hospitalization rather than incidence of severe illness.

Reviewer #3: Overall comments: 

This study examines a national data in Finland to investigate the association between household income and risk of COVID-19 infection and severe illness. A major strength of this manuscript is the database that they are using, which includes data on all individuals in Finland. In general, most of the manuscript is written appropriately. My main critique is that I think the analyses and writing need to be framed and much more rooted in modern causal epidemiologic terms. This involves thinking carefully about how adjustment is done and what the interpretation of each analysis. Many of the analyses seem to be focused on trying to understand mechanisms so formal mediation analyses may actually be the most appropriate. The strategy forward for doing this I think is conceptualizing the hypothesizing causal mechanisms using perhaps DAGs. This will help sharpen the exact questions being answered, how the analyses should be designed, and the appropriate interpretation of the results. Also, in general, I think the association with being foreign born came out very strongly but this relationship was not thoroughly explored.

Specific comments:

Intro

* The manuscript starts with the fact many studies have identified associations with low income and education and COVID-19, but they weren't adjusted for other covariates. I think an important question to ask is: Should they be adjusted? There are significant association between occupation, ethnicity, household size, and income. Other covariates are not necessarily confounders but may in fact be important mediators of effect. A simple example. Being foreign born leads to only able to acquire low income jobs leading to the need to live in multi-generational households, which leads to increased risk of COVID-19 due crowding. This manuscript (and others) by Camara Jones provides a lot insights in how to conceptualize design choices when considering disparities (https://academic.oup.com/aje/article/154/4/299/61900). It mostly applies to racism in the United States by a lot of the principles are applicable. In the end, these covariates are not independent biologic-like factors, but representative of a complex societal network and system that we all live in.

* The introduction is written okay, but I feel like it bounces around a lot. I get the sence that the questions of interest relate to household income and context in understanding COVID-19 transmission, but I do not come away understanding the specific questions or actual hypotheses. I think the focus of the first two paragraphs needs to be sharpened. The third paragraph is clear.

Methods:

* A major major strength of this paper is the data that is being used. It contain information on all individuals and households in Finland.

* I somewhat understand the rationale for only including households with working-age individuals, but in general the older population is the most concern for COVID-19. There still is something to be said about COVID risk in older households too. They are not likely isolated and inclusion could perhaps yield some insights about COVID acquisition through occupational hazards vs. non-occupational.

* I think more clarity on the rationale for selecting covariates and designing analyses are needed. In general, I would strongly advocate for drawing a DAG (directed acyclic graph) to formalize the hypothesized causal mechanisms at play. This also helps formalize how analyze should be design and how to handle adjustments.

* For example, when invoking mechanism I would think more about thinking about mediation analyses and trying to understand direct and indirect effects. This approach handles covariates differently then if one thinks they are confounders. I think the authors in general should consider carefully whether some of the analyses should be changed to formal mediation analyses (either causual mediation or standard mediation approaches would be appropriate).

* I am not sure I understand the rationale of the different models that add different covariates. Again, I would try to incorporate some more causal framing to designing the analyses. What question gets answered in each model? What questions get answered by comparing Model 2 to 3, for example. Is an unadjusted vs. adjusted analysis all that is needed? Again the question of whether covariates are confounders or mediators in a particular analysis is also not answered, and I think a major question here (or at least for interpretation).

* I would avoid the using "predict". These aren't really prediction models (which would be cross-validated to assess prediction performance) but rather identifying associations. I think also the word "effect" is used inappropriately throughout the results when most things are associations.

* Use of multi-level models is appropriate. It very much like the use of intraclass correlations for households.

Results

* I see in paragraph 2 of the results how the different models were interpreted. It is not very clear what it means to say that being foreign born "explained" the higher incidence. Again, I think the language needs to be framed in modern causal epidemiological terms and concepts like mediation. For example, the fact that association with income was attenuated with foreign background I think might suggest that income is a mediator of the impact of being foreign born (because the causal relationship can't go the other way) that one is mediating. If only interesting at looking at the impact of income, it is appropriate to adjust foreign born as a confounder too though. But this clarity on what the results mean does not come through.

* Rather than an interaction between income and household size…I would be most interested in income and foreign born interaction. I think this should be seriously considered.

* The thing that stands out the most to me in Figure 4 is how the relationships change once adjusting for foreign background. This to me suggests important aspects of the relationship between foreign background, income, and household size. Again assessing the interaction with being foreign born seems like it would be quite interesting.

* It would be interesting to understand the impact of neighborhood socioeconomic impacts as will. There are some basic descriptors of neighborhood (e.g., urban). Neighborhoods are often segregative into areas with great heterogeneity in terms of social vulnerability, but I don't these variables capture that.

* I would more clearly label the lowest and highest income quintiles throughout.

* What about clustering that happens at the neighborhood level and important neighborhood level covariates?

Discussion

* I think in general more contextualization is needed in terms of what the findings mean beyond the different analytic strategies (e.g., this analysis was adjusted this one was not). What is larger of importance of what the results mean because of these analytic differences.

* For example, I think more contextualization is needed in terms of this finding "In fact, the higher risk of severe COVID-19 in low-income households seems to be strongly driven by personal occupational status and a foreign background." Why would this be the case? What are the mechanisms

[LINK]

---

## [Decision Letter · Decision Letter 2]

5 Apr 2022

Dear Dr. Saarinen,

Thank you very much for submitting your manuscript "Income differences in COVID-19 incidence and severity in Finland: a population-based cohort study of individuals nested in households" (PMEDICINE-D-21-04207R2) for consideration at PLOS Medicine. 

Your paper was evaluated by a senior editor and discussed among all the editors here. It was also sent to independent reviewers, including a statistical reviewer. The reviews are appended at the bottom of this email and any accompanying reviewer attachments can be seen via the link below:

[LINK]

In light of these reviews, I am afraid that we will not be able to accept the manuscript for publication in the journal in its current form, but we would like to consider a revised version that addresses the reviewers' and editors' comments. Obviously we cannot make any decision about publication until we have seen the revised manuscript and your response, and we plan to seek re-review by one or more of the reviewers. 

We expect to receive your revised manuscript by Apr 26 2022 11:59PM. Please email us (plosmedicine@plos.org) if you have any questions or concerns.

We look forward to receiving your revised manuscript. 

Sincerely,

Beryne Odeny, 

PLOS Medicine

plosmedicine.org

Comments from the reviewers:

Reviewer #1: We thank the authors for addressing our previous concerns, as well as the point raised by Reviewer #3 on adjustment for other covariates. The manuscript appears clearer on the whole. However, in Table 2, The second "Model 5" might be "Model 6".

Reviewer #3: Overall comments: 

This study examines a national data in Finland to investigate the association between household income and risk of COVID-19 infection and severe illness. The authors made several changes which I think have strengthened the paper. In particular, examining the interaction between being foreign-born and income gradient. My main critique is that there still seems to be a bit of a disconnect between what the authors think is important to analyze with regards to socioeconomic drivers of COVID-19. In the intro and discussion, they talk about broader socioeconomic drivers, many of which could be the root causes of the disparities seen in COVID-19 outcomes (and also the reason there is an association with income and COVID-19). They also seem to discuss in essence income as a mediator of being foreign-born. However, the analysis focuses on the independent association of income with COVID-19, yet there isn't as much discussion about the important mechanisms of income that are independent of the other confounders. Thus, the focus on income when the results seem to point to the fact that there actually other more important drivers at play (being foreign-born in particular…and income as a mediator of that) is a bit confusing and should be sharpened and clarified. The reason to focus on income (and only income) in this setting needs to clear and compelling (currently I generally find the confounders more compelling and there is also more discussion of them as well).

Specific comments:

Intro

* "thus it remains unclear whether the worse outcomes reflect socioeconomic differences in incidence, case severity, or both" - I think this could be more specific by saying whether socioeconomic status influences risk of exposure and infection, outcomes once some is infected, or both. 

* Also, I think socioeconomic status often encompasses a broader array of factors including income, education, occupation, social class (which is also tied with race/ethnicity) (per APA definition). If this analysis is focused on income, I would use that specifically (which is also different than wealth).

* When reading the introduction (particularly second paragraph), I still get the sense that this analysis is focused on the broader milieu of socioeconomic factors as opposed to just income. The discussion of household clustering, occupation exposure seems to suggest that understanding their role is also important to this analysis (and then I would also add foreign-born status to that list). However, if the goal is only assess how income-level may drive infections and severity, I think the discussion should be focused there. Household size, occupation, race/ethnicity are then just confounders. How does income drive COVID things independent of household size, occupation, race/ethnicity? Is it related to underlying health access (which could lead to more comorbidities and then also worse COVID outcomes)? Access to information? In general (and I imagine particularly in Finland where there is a strong public healthcare system), the financial component is less the issue, but income may be an important proxy for several other risk factors (as the authors note).

Methods:

* The authors state they want to focus in on household level factors, but neighborhood-level factors are also a likely important confounder to consider when considering the relationship between income and COVID outcomes. If available, I would consider how it could be incorporated (assessments of clustering at the neighborhood level are also still interesting). If unavailable, that is fine, but I wasn't convinced by the rationale for not including them. Ultimately, they likely represent important confounders, and if unavailable, should be included as a limitation. 

* I don't think necessary to say "in response to peer review comments". I think can just say "We conducted three sensitivity analyses."

* 

* For under vs. over 65 year old households, the definitions are not clear. Were only households with everyone under 65 included? Where were households where the main earner was less than 65, but some over 65 years olds also lived (e.g., mixed households with grandparents)? I understand excluding households where everyone is over 65 years old, but this latter group seems like they should still be included (and perhaps they are). I think the definitions need to be clarified and also needs clarity of where these mixed households fit in. This composition is even an interesting covariate.

Results

* Incidence models - I would create a separate sentence highlighting that when foreign background was included as a covariate, the association of income and COVID-19 was largely attenuated. As presented, seems like a secondary point. 

* Is it possible to present a threeway interaction between income, foreign-born, and household size. When I see the interaction between household size and income, the first question that I have…are foreign born individuals more likely to have larger household sizes. And then the household size interaction ends up still being related to foreign born status too. This may or may not be true but I think a question that should be answered in someway. If space is an issue, could consider only included the adjusted model in the figures and also figure 6 and 7 could potentially be supplementary.

* Even without a mediation analysis, it is clear that income is an important mediator of being foreign born it seems (agree that there aren't measured mediators of the effect of income on COVID). But it does not mediate the full effect as even among the highest income, there are differences between native Finnish and foreign-born. This piece gets lost for me. 

Discussion

* I think leading with this sentence "Individuals living in low-income households appeared to have higher odds of both COVID-19 incidence and severity" ends up being somewhat misleading (even though it is clarified later). I think needs to also include that is specific for foreign born individuals only and not among native Finnish. The point is there is an income gradient among foreign born and there is not among native Finnish.

* 

* I also don't know that I would say that this relationship is largely confounded by foreign born. Though technically true, that implies there was a false association (which I don't think is the authors' point). I think I would try to clarify what the results indicate, which to me is that being foreign-born is a major driver of differences in outcomes, and the income mediates some but not all of it (and also that there is no income gradient among native Finnish).

* The response to reviewers and analysis seems to emphasize that the point of this analysis is to assess the association of income with COVID-19 outcomes that is independent of other socioeconomic factors such as household size, occupation, foreign-born, etc. However, the intro nor the discussion really dives into the important independent mechanisms of income. Most of the discussion actually seems to discuss income as a mediator for these other more proximal variables. This is the basis for my previous comments that it might be prudent to also have some focus on income as mediator for these more proximal variables (as opposed to mediators of income itself). But if the focus is on the independent effects of income, there needs to be discussion of those specifically. 

* To be fair, my thinking generally is in line with the authors framing of the broader socioeconomic milieu being very important drivers, with income as one of the mediators of them. But the stated focus of the analysis is somewhat contrary to that, so that discussion needs to be there.

* 

* For example, "Taken together, our results suggest that income is an independent risk factor for COVID-19 incidence only among people with foreign background. The disadvantage among those with foreign background and low household income may relate to the work and school exposures, household size, and comorbidities that we were able to measure." These sentences combine income as mediator of foreign background (along with household composistion, work, school) and the independent mechanisms of income alone. The discussion of how income may independently relate to COVID-19 outcomes is lacking. 

* "Foreign background, in turn, may capture ethnic differences in the risk for severe COVID-19, the reasons for which are likely to be multifaceted and currently not very well understood". I am not sure I agree with this sentence. First, it does seem to imply a biologic reason for differences (which I am not sure the authors intended). Second, I do think we know a lot about why being foreign-born person relates to COVID-19, many of which the authors do discuss throughout. But it relates to structural racism, issues with access (which could be driven by language, education, access to information), socioeconomic factors, neighborhood deprivation, etc. There has been a lot of literature about drivers of health disparities (even prior to COVID), and many probably are also relevant to Finland. This discussion could be more robust and perhaps combined with the discussion in the preceding paragraph.

[LINK]

---

## [Decision Letter · Decision Letter 3]

10 May 2022

Dear Dr. Saarinen,

Thank you very much for re-submitting your manuscript "Income differences in COVID-19 incidence and severity in Finland: a population-based cohort study of individuals nested in households" (PMEDICINE-D-21-04207R3) for review by PLOS Medicine.

I have discussed the paper with my colleagues and the academic editor and it was also seen again by one reviewer. I am pleased to say that provided the remaining editorial and production issues are dealt with we are planning to accept the paper for publication in the journal.

The remaining issues that need to be addressed are listed at the end of this email. Any accompanying reviewer attachments can be seen via the link below. Please carefully consider the reviewers outstanding concerns and take these into account before resubmitting your manuscript:

[LINK]

We look forward to receiving the revised manuscript by May 17 2022 11:59PM.   

Sincerely,

Beryne Odeny, 

PLOS Medicine

plosmedicine.org

Requests from Editors:

1) Please revise your title to include “foreign background,” considering the association between living in low-income households and COVID-19 incidence and severity is specific for foreign born individuals only and not among native Finnish. 

2) Please revise your subtitle to “…nested within households” as opposed to “…nested in households.”

3) Abstract and Results- Please include the actual amounts or percentages of relevant outcomes in addition to the ORs

4) Discussion, Line 457 - Please avoid assertions of primacy (“Our study is the first to..."). Instead, state “To our knowledge, …” or similar

5) Please include definitions for all abbreviations in the tables, e.g., ICC

Comments from Reviewers:

Reviewer #3: Overall comments: 

I thank the authors' for their efforts in addressing my comments, and I think they have provided additional clarity on several issues that were raised. There are still a few points that I think would strengthen the paper (which I list below), though I think there is likely a conceptual disagreement on how I and the authors' think they should be approached (which is okay). I only raise them again as I think causal and conceptual clarity is particularly important in discussions of disparities—in large part because of the history how of they have been inadequately discussed—but would defer to the editors and authors' on how best to handle them at this stage.

Specific comments:

* I think there is too much interchangeability between the use of socioeconomic factors and income in the introduction. As mentioned previously, I think at least some discussion that clearly differentiates the mechanisms between the two is important somewhere. They do so in discussion of the methods, though some readers may interpret a false equivalence between the two after reading the intro. I do not think the authors' intend this, but do think it is easy to miss the distinction, which is key to this paper, because they have too often been conflated for each other in the past.

* I think the threeway interaction between income, foreign-born, and household size could still be interesting and important. It is difficult to assess just from the p-values since it is not unexpected that p-values would be high in a threeway interaction (p=0.29 for a threeway is not too high). When point estimates and CIs are graphed, are there any insights from the trends…is everything still driven by being foreign born? That would be the data that would inform my assessment of whether it is relevant or not. It would be great if authors' could share these results at least and could consider as supplemental figure. Again, the authors' may already done this type of full assessment, but from the reponse it seems solely based on the p-value.

*

* It took me a few reads to understand the point being made in the discussion by "in the total population, individuals living in low-income households had higher risk of both COVID-19 incidence and severity…" My first two reads interpreted this has trying to make a claim about some level of generalizability of the findings (and not in just an artefactual mathematical sense) even though the following statements did clarify the issue. Though I think the authors' are trying to say that they found a similar result to previous studies, but they also find that it is in fact not generazable. Could consider saying something like "Prior evidence has shown associations with income in the total population, but our deeper dive indicated that these trends were in fact driven primarily by a specific population and not generalizable to the whole population" or something of that nature.

[LINK]

---

## [Decision Letter · Decision Letter 4]

23 May 2022

Dear Dr. Saarinen,

Thank you very much for re-submitting your manuscript "Income differences in COVID-19 incidence and severity in Finland among people with foreign and native background: a population-based cohort study of individuals nested in households" (PMEDICINE-D-21-04207R4) for review by PLOS Medicine.

I have discussed the paper with my colleagues and the academic editor and it was also seen again by ome reviewers. I am pleased to say that provided the remaining editorial and production issues are dealt with we are planning to accept the paper for publication in the journal.

[LINK]

We look forward to receiving the revised manuscript by May 30 2022 11:59PM.   

Sincerely,

Beryne Odeny,

PLOS Medicine

plosmedicine.org

Comments from Reviewers:

Reviewer #3: Thank you to the authors' for their revisions and their inclusion of a figure of the threeway interaction. I do think it really helps to emphasize the studies main findings. I appreciate the authors' clarification that they are using income as a proxy for the multidimensional construct for socioeconomic status. I think clarifications are needed in two places: in the intro and strengths and limitations. 

When the authors say that household income is a "reliable" measure, do they mean reliable because it was available for everyone or reliable because it is an adequate proxy for this complicated multidimensional construct. It reads like the latter, but I think the results themselves directly speak against it being the latter. I would change the wording of this in the intro.

I also think one or two sentences need to be added to the strengths and limitations to be clear that this is a limitation (currently the wording seems to describe this as a strength). Specifically, that they used income as a proxy for a complicated multidimensional construct, which is not ideal because socioeconomic status also includes considerations related to race, community, etc., but was done because that data was what was reliably available. The completeness of the income data is a strength but using at is a proxy for complex construct is certainly an important limitation.

[LINK]

---

## [Decision Letter · Decision Letter 5]

31 May 2022

Dear Dr Saarinen, 

On behalf of my colleagues and the Academic Editor, Dr. Aaloke Mody, I am pleased to inform you that we have agreed to publish your manuscript "Income differences in COVID-19 incidence and severity in Finland among people with foreign and native background: a population-based cohort study of individuals nested within households" (PMEDICINE-D-21-04207R5) in PLOS Medicine.

PRESS

Sincerely, 

Beryne Odeny 

PLOS Medicine